# Super Potent Bispecific Llama VHH Antibodies Neutralize HIV via a Combination of gp41 and gp120 Epitopes

**DOI:** 10.3390/antib8020038

**Published:** 2019-06-18

**Authors:** Nika M. Strokappe, Miriam Hock, Lucy Rutten, Laura E. Mccoy, Jaap W. Back, Christophe Caillat, Matthias Haffke, Robin A. Weiss, Winfried Weissenhorn, Theo Verrips

**Affiliations:** 1Department of Biology, Faculty of Sciences, Utrecht University, 3584 CH Utrecht, The Netherlands; n.m.strokappe@gmail.com (N.M.S.); lucy.rutten@gmail.com (L.R.); 2QVQ Holding bv, Yalelaan 1, 3584 CL Utrecht, The Netherlands; 3Institute de Biologie Structurale (IBS), CNRS, CEA, Université Grenoble Alpes, F-38000 Grenoble, France; Miriam.hock@googlemail.com (M.H.); christophe.caillat@ibs.fr (C.C.); winfried.weissenhorn@ibs.fr (W.W.); 4Immunocore Ltd., 101 Park Drive, Milto, Abingdon OX14 4RY, UK; 5Division of Infection and Immunity, University College London, London WC1E 6BT, UK; l.mccoy@ucl.ac.uk (L.E.M.); r.weiss@ucl.ac.uk (R.A.W.); 6Pepscan B.V., Zuidersluisweg 2, 8243 RC Lelystad, The Netherlands; jjaapwillemback@eurofins.com; 7European Molecular Biology Laboratory, Grenoble Outstation, 6 rue Jules Horowitz, 38042 Grenoble, France; Matthias.haffke@novartis.com; 8Global Discovery Chemistry, Novartis Institutes for BioMedical Research, Novartis Pharma AG, Novartis Campus, 4002 Basel, Switzerland

**Keywords:** Aids, HIV, Llama Antibodies, bi-specific VHH, pepscan, competition studies, epitope mapping, co-crystallisation

## Abstract

Broad and potent neutralizing llama single domain antibodies (VHH) against HIV-1 targeting the CD4 binding site (CD4bs) have previously been isolated upon llama immunization. Here we describe the epitopes of three additional VHH groups selected from phage libraries. The 2E7 group binds to a new linear epitope in the first heptad repeat of gp41 that is only exposed in the fusion-intermediate conformation. The 1B5 group competes with co-receptor binding and the 1F10 group interacts with the crown of the gp120 V3 loop, occluded in native Env. We present biophysical and structural details on the 2E7 interaction with gp41. In order to further increase breadth and potency, we constructed bi-specific VHH. The combination of CD4bs VHH (J3/3E3) with 2E7 group VHH enhanced strain-specific neutralization with potencies up to 1400-fold higher than the mixture of the individual VHHs. Thus, these new bivalent VHH are potent new tools to develop therapeutic approaches or microbicide intervention.

## 1. Introduction

AIDS remains one of the largest global health problems and annually an estimated 1.8 million people die an AIDS related death. While antiretroviral therapy and careful clinical management can render HIV a chronic disease, the treatment is expensive and has many adverse effects [1]. Thus in the absence of a vaccine, prophylactic therapies to prevent HIV-1 infection are urgently needed. The encouraging results of an antiretroviral-based gel microbicide [2] suggest that microbicidal prevention methods merit investigation. Recently, many broad and potent HIV neutralizing monoclonal antibodies (mAb) have been discovered [3,4,5,6,7,8,9,10,11]. However, while monoclonal antibody (mAb)-based microbicides have been evaluated before [12], the application of this is limited by their expensive production and the need for cold distribution system. The variable domains of the heavy chain of heavy chain only antibodies (VHH) derived from llamas or other *Camelidae* may be better alternatives, as they can be produced relatively cheaply in microorganisms like bacteria or yeast [13] and are often stable at high temperatures [14]. This is predominantly related to their small size, which is a 10-fold smaller than that of a conventional antibody. Moreover, their small size and more than average length of CDR 3 allows them to bind to recessed epitopes, like the CD4 binding site (CD4bs) of HIV-1.

Previous immunizations of llamas (*Lama glama*) have generated anti-HIV-1 VHH specific for the CD4bs on gp120 or gp140 [15,16,17] with the most potent and broadest VHH, J3, neutralizing 96% of the HIV-1 strains tested from subtypes A, B, C, D, G and circulating recombinant forms AC, ACD, AE, AG, and BC [16]. A mix of these VHH together with J3 was tested in vitro and showed to be neutralizing as potently as any of the individual VHH, and a 100% coverage for the panel of 60 viruses tested was predicted. Similarly, combinations of antibodies that bind independent epitopes have also been shown to neutralize in vitro as efficiently mixed together as when used in isolation [18].

A number of broadly neutralizing VHH, with breadths up to 82%, targeting epitopes on the HIV envelope glycoprotein (Env) other than the CD4bs, e.g., the co-receptor binding site or gp41, have been isolated as well, but their epitopes had not been defined [17,19,20]. Some of these VHH are able to neutralize the few strains that are not neutralized by J3 or by 3E3, another VHH obtained from our immunized llamas, which neutralizes 80% of virus strains [21]. By using a mix of VHH targeting the CD4bs and VHH targeting other epitopes, broader neutralization is expected. An additional advantage of the use of a mix of VHH is that it reduces the chance of the emergence of escape mutants. This has been observed in studies in which a combination of some broadly neutralizing conventional human antibodies targeting independent epitopes were passively transferred to humanized mice [3,4]. Using covalently linked VHH often has an advantage over using monovalent VHH or a mix of VHH, as the bivalent VHH may have higher potencies than the constituent monovalent VHH, due to enhanced avidity [22,23]. Furthermore, HIV-1 neutralization potency can be enhanced by heteroligation of two distinct single-chain Fv (scFv), into bi-specific molecules [24]. In contrast, bi-specific VHH targeting the CD4bs as well as the co-receptor binding site did not show any increase in potency, but an extended breadth was confirmed. Nevertheless bi-specific VHH may have an advantage when used in gene therapy in a vectored-immunoprophylaxis (VIP) adeno-associated virus (AAV) [25], because the size of a bi-specific VHH genes do not exceed the maximal allowed size of the transgene insert.

Therefore, we aimed to design and produce bi-specific anti-HIV-1 VHH, which have neutralization abilities superior to those of the two best VHH, J3 and 3E3, regarding breadth and potency, for various applications. In order to design these molecules, i.e., to make the best combinations, we first determined the epitopes of several VHH. VHH that bound to epitopes other than the CD4bs were linked to J3 and 3E3 and most of these bi-specific VHH have improved breadth and potency against certain viral strains, compared to the equimolar mix of the constituent VHH.

## 2. Results

### 2.1. Competition-Based Determination of 4 Different Epitope Groups

Previously, we have selected over 100 different HIV-1 neutralizing VHH starting with various immune libraries (Most important data summarized in Figure 1 and Figure 2) [15,16,17,21,26]. To be able to make bi-specific VHH, two VHH that target independent epitopes need to be linked together. Preliminary evidence indicated that a number of the selected VHH do not compete with J3 or 3E3 [19]. To determine the epitopes of the preselected VHH we performed a (cross-) competition assay in which seven of these VHH (3E3, J3, 1B5, 2E7, 1F10, 11F1F, 1H9) were tested for their competition against each other and against six other VHH that neutralize HIV-1 (Figure 3A). In the assay, Env was immobilized and a total of 13 neutralizing and an irrelevant VHH were added in excess to the plate. Following incubation, seven biotinylated VHH of interest were added and their binding was detected via peroxidase-conjugated streptavidin (Figure 3A), showing the competition against gp140UG37 (and Appendix A showing the competition against gp120IIIB and gp140CN54). These competition assays enabled the clear division into four groups of VHH targeting different epitopes. The first group is comprised of phylogenetically diverse VHH (Figure 2) targeting the CD4bs, which includes J3, 3E3, A12, D7 and C8. The second group includes phylogenetically related 2E7 and 11F1F and unrelated VHH 11F1B and 1E2. The third group of VHH, that probably block the co-receptor binding site, is composed of phylogenetically related 1B5, 1H9 and 2B4F and the unrelated 4D4. The fourth group contains one VHH, 1F10, which does not fully compete with any of the other VHH, indicating that it binds to a separate epitope. To obtain more insight into the location of the different epitopes on Env, a competition assay was performed with these VHH versus b12 (a bnAb targeting the CD4 binding site), 17b (a bnAb targeting the co-receptor binding site), sCD4 and anti-gp41 MPER bnAbs (2F5, 4E10) were performed (Figure 3B). This revealed that group I targets the CD4 binding site, group II targets gp41 independent of the MPER epitope, group III targets the co-receptor binding site and group IV also seems to target the co-receptor site (Figure 3B). Since this competition was only performed in one way, differences in affinity between the VHH and the mAb may have led to false negatives. The epitopes recognized by the VHH were subsequently further characterized by pepscan analyses and structural studies.

#### 2.1.1. Epitope Group I, VHH That Bind to CD4bs

This group consists of J3, 3E3, A12, C8, D7 and their family members. The VHH were obtained directly from the phage library or selective elution with sCD4, of phages carrying these VHH. Moreover, previous work on competition experiments of these VHH, with sCD4 and b12, pointed out that this group of VHH is targeting the CD4bs [19]. During these studies it became clear that J3 as well as 3E3 enhance binding of 17b by approximately 3- or 4-fold, respectively. Although J3 and 3E3 are phylogenetically not related and were selected from different llamas, they have several features in common. For example, they share the very rare feature that they lack three residues in the CDR2 relative to their respective germ line [27]. The reinsertion of these three residues in J3 or 3E3 abrogates the binding abilities of either VHH (data not shown and McCoy et al. [16]). Unlike e.g., VRC01, which also binds to the CD4bs, J3 and 3E3 do not bind to RSC3, a resurfaced gp120 molecule in which there are many alterations outside the structurally invariant part of the CD4bs, including in the bridging sheets [9]. These data suggest that that J3 and 3E3, like CD4, bind to the bridging sheets of gp120. Moreover, J3 in particular shows unusual degrees of maturation outside of its paratope, although less than bnAb selected from humans. The foot print of J3 and 3E3 were determined by modeling [27]. The most characteristic feature of the binding of J3 to gp120 is that its foot print is very similar to the foot print of CD4. It was predicted by the model that Tyr99_J3_ occupies a position similar to Phe43_CD4_ resulting in a similar hydrophobic interaction with gp120. Recently these interactions were confirmed by co-crystallization [manuscript in preparation]. The competition assay shows that sCD4 competes with J3, 3E3, A12 and C8 for binding to Env. However, A12 and C8 also compete with 17b, indicating that the interaction surface of A12 is also outside the CD4bs, which is less conserved than the CD4bs itself. This may be an explanation for the relatively low breadth of A12, which neutralized 42% of the viruses tested.

#### 2.1.2. Epitope Group II Consisting of 2E7, 11F1F, 1E2 and 11F1B and Their Family Members

2E7 and 11F1F share 91% DNA and 89% protein sequence identity and neutralize 21 out of 26 (81%) and 36 out of 45 (76%) viruses tested, respectively. While the potencies of 2E7 and 11F1F (median IC50 of 19 and 21 μg/mL respectively) are much weaker than those of the CD4bs VHH 3E3 and J3, these VHH can neutralize some viral strains (e.g., Du172.17 and CAP45.2.00.G3) that are resistant to either 3E3 or J3 (Appendix A). Based on the binding and competition experiments described above it is clear that 2E7, 11F1F, 1E2 and 11F1B bind to gp41 outside the MPER epitope as they don’t compete with bnAbs 2F5 and 4E10. Furthermore, no competition was observed with gp41 Heptad Repeat 1 (HR1) Abs HK20 and 3D6, that target amino acids 535–581 [8] and the immunodominant area (amino acids 579–613) [28] of gp41, respectively. Inhibition of 2E7 and 11F1F binding was seen in the presence of antibody 246-D (epitope sequence 590-QQLLGIWG-597 with the epitope core being LLGI), suggesting that these VHH bind to the C-terminal part of the HR1 region of gp41, but not to the epitope of 3D6, which is 599-SGKLICTTA-607.

This region of gp41 has been characterized as being immunodominant, thus it is not surprising that immunization of llamas with recombinant gp140 yielded antibodies targeting this region. However, it is novel that the VHH antibodies elicited can not only neutralize HIV, but do so with a breadth ranging across many subtypes. Therefore, the epitopes of 2E7, 11F1F, 1E2 and 11F1B were investigated by measuring their binding to overlapping linear and cyclic 15-mer peptides covering gp160 proteins derived from subtype A, B, C and CRF BC viruses (strains UG037, HXB2, SF162, ZM96 and CN54). All VHH showed binding to a peptide containing the sequence AVERYLKDQQLLGIWG (corresponding to residues 582–597 in HXB2 numbering, data not shown). Therefore, we focused on the broadest VHH, 2E7 in the following analysis. A peptide containing the epitope of 2E7 was used as a seed for a library in which each amino acid was uniquely substituted with Ala, Arg, Glu, Phe, Gly or Trp to obtain a limited substitution mutagenesis scanning. Clear binding peaks were observed corresponding to the peptides indicated in Figure 4, and were largely consistent for all tested subtypes represented in the peptide sets. The consensus sequence for the 2E7 epitope is (I/V)ERYL(R/K)DQQL (583–592). The limited substitution mutagenesis screening was in good agreement with the results obtained in the initial gp160 screening, showing that 2E7 is particularly sensitive to mutations in residues K/R588, D589, and L592.

Gp41 residues 582 to 596 adopt a helical conformation in the native gp140 structure and form the C-terminal part of the HR1 triple stranded coiled coil, which is hidden within the trimer interface [29]. We therefore tested binding of 2E7 to a fusion intermediate conformation of gp41 (gp41_int_) that contains part of HR1 including the 2E7 epitope fused in frame to the pIIGCN4 triple stranded coiled coil [30]. SPR measurements revealed a K_D_ of 0.592 nM (k(on): 9.52 × 10^5^ (1/Ms); k(off): 5.64 × 10^4^ (1/s) (Figure 5A) corroborating the interaction of 2E7 with activated gp41 that has the HR1 coiled coil exposed similar to the mode of action described for HR1 antibodies D5 and HK20 [31,32]. We next solved the crystal structure of 2E7 in complex with the peptide 582-AVERYLKDQQLLGIW-596 to a resolution of 2.9 Å (Table 1). The structure shows that the VHH interacts with the gp41 peptide in an unusual way. The gp41 helix packs lateral to one side of the VHH beta sheet (Figure 5B). The major contacts are hydrogen bonds between gp41 D589 and the hydroxyl and backbone amide of the CDR 2 residue T53 as well as gp41 W596 and framework residue T50. A salt bridge from gp41 K588 to CDR1 D32 (Figure 5B) and hydrophobic contacts of gp41 L592 to CDR1 A33 and CDR2 I51, and gp41 I595 and W596 to framework P47 further stabilize the interaction. Cα super-positioning of the 2E7-gp41 peptide structure and the native gp140 structure confirms that 2E7 would not be able to access the HR1 epitope which is hidden in the native Env conformation (Figure 6). We conclude that 2E7 targets the HR1 coiled coil, which is temporarily exposed during membrane fusion before refolding into the six helical bundle post fusion conformation [33].

#### 2.1.3. Epitope Group III Consisting of 1B5 and 1H9 and Family Members Recognize Part of the Co-Receptor Binding Site

Figure 3A shows that VHH 1B5 and 1H9 show a similar cross-competition pattern, moreover they compete with each other, indicating that they target a similar epitope. They compete with 17b, but not with b12 or sCD4 (Figure 3B). 1B5 and 1H9 were subjected to pepscan analysis, but neither bound to any of the peptides of the arrays of overlapping linear and cyclic 15-mer peptides covering gp160 proteins derived from various subtypes (data not shown). Escape mutant studies indicated that residues P417 and R419 [34] are involved in the interaction of 1B5 with gp120. We this suggest that group 3 VHH target the coreceptor site largely based on the competition assays (Figure 3B). The proposed locations of the 1B5 and 1H9 epitopes are shown in Figure 6, in which the residues P417 and R419 are shown in magenta.

#### 2.1.4. Epitope Group IV Consisting of 1F10 Binds to the Crown of the V3 Loop

1F10 neutralizes 18 out of the 26 viruses tested (69%). Based on the competition experiments it is clear that 1F10 competes with 17b, but not with sCD4 or b12. It competes marginally with all the other VHHs except for the CD4bs targeting VHH J3 and 3E3 (Figure 3 and Appendix A). Furthermore, 1F10 does not compete with CD4bs antibody HJ16 either [8], supporting that this VHH binds a non-CD4bs epitope. However, 1F10 does compete with HGN194, a neutralizing Ab which binds to the crown of the V3 loop. Pepscan analysis with 1F10 was performed on subtype A, C and CRF BC viruses (strains UG037, ZM96 and CN54). Clear binding peaks were observed for peptides derived from the V3 region and were largely consistent between the three subtypes. (data not shown). The consensus sequence for the 1F10 epitope is IRIGPGQT (307–314 according to the HXB2 numbering) which overlaps with the HGN194 epitope RRSVRIGPGQTF (304–315). Single amino acid full replacement analyses were performed using a cyclic peptide (CRSVRIGPGQTFYAC) and two linear peptides (KRIRIGPGQTFY and KSINIGPGRAFA), each containing the sequence region recognized by 1F10. The replacement analysis identified the core epitope of 1F10 as IxIGPGxT (Figure 4B). The epitope of 1F10 is depicted in Figure 6A, where it is highlighted in green.

### 2.2. Construction of Bispecific anti-HIV VHH

Of the four CD4bs (group 1) VHH, J3 and 3E3 have the broadest and most potent HIV neutralization ability (Appendix A). Thus, these two VHH were chosen for the construction of bi-specific molecules. VHH targeting distinct epitopes were paired with either 3E3 or J3 to maximize breadth. From group 2, VHH 2E7 and 11F1F were chosen and 1H9 is representing group 3 as it neutralizes CAP45.2.00.G3, a strain not neutralized by J3, and because its median IC50 is lower than that of 1B5 (5 µg/mL versus 13 µg/mL). The sole representative of group 4, VHH 1F10 (targeting the V3 loop), was also chosen as candidate for bi-specific VHH as it is broad and neutralizes at least two of the viruses (e.g., Du172.17 and CAP45.2.00.G3) J3 does not [16,17]. An overview of the neutralization IC_50_ values for these VHH is given in Appendix A.

VHH targeting the CD4bs were joined with VHH targeting the HR1 of gp41, the co-receptor binding site or the V3 loop (Figure 6) by a flexible 15 or 25 amino acid glycine-serine (GGGGS) repeat linker to form bi-specific VHH molecules. As the 15 and 25 linkers did not show a significant difference in their functionality (data not shown), the 15 linkers were chosen for further investigation. The bi-specific VHH with the CD4bs VHH on the N-terminal side seem to behave superior to those with the CD4bs VHH on the C-terminal side (data not shown). The bi-specific VHH bind to all tested Env proteins if either constituent VHH is able to bind (Figure 3C).

In general, the ability of the bi-specific VHH to compete with human bnAbs is equal to the combined competing abilities of both monomeric components. However, for bi-specific constructs containing 1F10, the ability to compete with 17b is reduced (J3-1F10) or turned into enhanced binding (3E3-1F10). This may be due to conformational changes induced by J3 and 3E3, as both individual VHH enhance 17b binding. Reduced competition with 17b is also seen with 3E3-1H9.

### 2.3. Broad and Potent HIV Neutralization by bi-Specific VHH Targeting a Combination of gp120 and gp41 Epitopes

Preliminary neutralization experiments indicated that the bi-specific constructs containing 1H9 did not yield a large improvement of breadth or potency and thus it was not characterized further. The remaining bi-specific VHH were tested for the ability to neutralize viruses resistant to either J3 or 3E3 to test whether the breadth of the bivalent construct was higher than that of J3 and 3E3 alone. Du172.17, a subtype C tier 3 virus, is not neutralized by J3, but is neutralized by all three J3-containing bi-specific VHH with IC50 values equal or lower than those of 1F10, 2E7 or 11F1F alone (Figure 7A). The same holds true for TV1.2, this subtype C tier 2 virus, is resistant to 3E3, but is neutralized by all three 3E3-containing bi-specific VHH or their mixes (Figure 7B). However, for some combinations the mixed monomers appear to be more potent than the linked constructs, therefore an additional six viruses from different subtypes and tiers were tested (Figure 7C). These viruses were in some cases susceptible to neutralization by the mix of both VHH constituting the bi-specific VHH, however, neutralization at least as potent as that of the most potent component VHH was seen for both linked and unlinked VHH mixtures in all cases (data not shown). Dependent on the HIV strain, some bi-specific VHH were substantially more potent than the equimolar mix of the constituent VHH. The bi-specific VHH containing 1F10 did not show greatly enhanced potencies compared to the equimolar mixes of the constituent VHH for any of the strains tested, whereas the bi-specific VHH with 2E7 and 11F1F have greatly enhanced potencies up to a 1400-fold, especially on the two C-clade viruses. For the subtype A and B viruses, approximately equivalent potency was seen for all bi-specific VHH compared to their respective unlinked equimolar mixtures, only J3-11F1F showed an increase in potency of a factor 6.3 against MS208.A1 and up to a 5-fold potency increase for Bal.26. In contrast, large increases in potency for the bi-specific VHH relative to the unlinked equimolar mixes were observed against the two subtype C viruses. A particularly large potency increase was seen for 3E3/J3-11F1F and J3-2E7 against the tier 2 ZM214M.PL15 virus and 3E3/J3-11F1F against the tier 1 96ZM651.02 virus. We conclude that the increase in breadth and potency was most efficient against clade C with a combination of anti-CD4bs VHH and anti-gp41 VHH.

## 3. Discussion

To broaden neutralization capacities, but also to reduce the chance of the emergence of escape mutants, targeting two independent epitopes on Env is likely to be beneficial [35]. Recently, bi-specific antibodies, targeting amongst others, the CD4bs [36,37] and MPER [38,39] showed broad and potent neutralization. Furthermore, fusions of CD4 and mAb 17b revealed synergetic effects [36,40] as well as VHH fusions revealed important synergistic effects [22,40,41]. To determine optimal combinations for the construction of bi-specific VHH we first determined and characterized a number of new VHH, which could be combined with the most potent neutralizing VHH J3 or 3E3 [16,21]. 13 new VHH were classified into four groups recognizing non-overlapping epitopes. Only for 1F10 was some competition seen with the epitopes of other VHH (1B5/1H9 and 2E7/11F1F). This competition is most likely due to steric hindrance or because of conformational changes occurring after binding of one of the VHH, preventing the binding of the other.

To determine the epitopes of the VHH binding to the three epitope groups other than the CD4bs in more detail, we subjected the VHH of these groups to pepscan analysis. 2E7, as a representative of epitope group 2, binds to gp41. It binds to the linear sequence (I/V)ERYL(R/K)DQQLLG(L/I)W at position 583–596 according to the HXB2 numbering. Binding to the predicted gp41 epitope was confirmed by the crystal structure of 2E7 in complex with a peptide of this epitope. The epitope is part of the C-terminal part of the HR1 coiled coil that stabilizes the trimer interface in the native gp140 conformation [29,42,43]. In this native like conformation, the epitope is inaccessible to 2E7 due to steric clashes. Instead we show that 2E7 binds with high nanomolar affinity to the fusion intermediate conformation of gp41 that bridges the viral and cellular membranes during the initial step of membrane fusion [33] and is occluded by HR2 in the gp41 post fusion conformation [44]. Its mode of action is thus similar to HR1 mAbs D5, HK20 and 8066 whose epitope is located at the N-terminal part of HR1 [31,32,45]. Although these mAbs neutralize HIV-1, their breadth and potency is largely increased when they are used as smaller Fabs or single chain antibodies [31,32,45], indicating that the site is difficult to access for a conventional antibody during the fusion reaction. It therefore remains to be tested whether 2E7 has the same breadth and potency as a complete Fc containing antibody.

1B5 and 1H9, (epitope group 3) compete with 17b, an Ab that targets the co-receptor site. This in combination with preliminary data of co-crystallization experiments [manuscript in preparation] suggests an epitope overlapping with the co-receptor binding site.

High resolution pepscan analysis revealed that 1F10 (epitope group 4) binds to the sequence IRIGPGQT (HXB2 position 307–314) in the crown of the V3 loop, an epitope also targeted by the human mAbs 447-52D and HGN194. The latter is able to prevent the mucosal transmission of a subtype C SHIV when passively transferred [46]. The VHH described here thus target the following four distinct epitopes: the CD4bs, cluster1/HR1 of gp41, the co-receptor binding site or the V3 loop. All of the VHH epitopes are depicted in Figure 6A.

Bi-specific antibodies are currently investigated for a number of purposes and in general the bi-specific antibodies are performing quite well in vitro as is demonstrated by Asokan et al. [36]. In particular the construct VRC07-PG9-16 performed very well by neutralizing up 97% of the viruses tested with a median potency of 0.055 μg/mL. However, the situation in vivo is more complicated as there is a reasonably high chance that the bi-specific antibody initiates an immune response. Results of clinical trials of Ablynx show that VHH have a low risk for triggering immune responses in humans, because of their physical-chemical properties, the folding of CDR1 and 2 and their small size.

We constructed bi-specific VHH, containing either J3 or 3E3 (epitope group 1) in combination with a VHH that binds to an independent epitope. The constructs containing 1H9, did not show enhanced neutralization compared to the mix of the constituent VHH in preliminary experiments, so therefore they were not studied further. This observation agrees well with the fact that other bi-specific VHH targeting the CD4bs and the co-receptor binding site simultaneously did not show enhanced potency [20]. The increases in potency of the bi-specific VHH containing 1F10 were not spectacular for the viruses that were neutralized by either constituent VHH. This suggests that 1F10 hardly binds simultaneously with J3 or 3E3. The bi-specific VHH with the CD4bs VHH on the N-terminal side neutralize superior to those with the CD4bs VHH on the C-terminal side. The largest improvements in potencies, up to 1400-fold, are obtained with the constructs containing the gp41 targeting 2E7 or 11F1F as counterparts. The improvements are highest towards the viruses 96ZM651.02 and ZM214, both C-clade viruses, whereas towards the viruses from other clades, the improvements are less than 10-fold. Unexpectedly, J3-11F1F as well as 3E3-11F1F showed more than 200-fold increased potencies towards ZM214, whereas monomeric 11F1F was unable to neutralize this virus at a concentration of less than 1 µM. A plausible explanation for this may be that the binding of J3 or 3E3 causes conformational changes that allow 11F1F to bind or enhance 11F1F binding, confirming an epitope present on the intermediate conformation of HR1.

## 4. Methods

### 4.1. Materials and Methods

#### 4.1.1. Proteins

Monoclonal antibodies b12 (EVA3065, by Dr D. P. Burton [47], 17b (ARP3071 by Dr J. Robinson) [48], 2F5 (EVA 3063, by Dr H. Katinger), 4E10 (ARP3239, by Polymun), and L120 ARP359, by Becton Dickinson) the recombinant proteins gp120IIIB (EVA607, by ImmunoDiagnostics), gp140UG37 (ARP698, by Polymun), gp140CN54 (ARP699 by Polymun), gp41 (ARP680) and human soluble CD4 (EVA609, by ImmunoDiagnostics) were obtained through the Centralized Facility for AIDS Reagents (CFAR), the National Institute for Biological Standards and Controls (NIBSC).

#### 4.1.2. Viruses

Replication competent virus stocks were prepared from HIV-1 molecular clones by transfection of 293T cells. HIV-1 envelope pseudotyped viruses were produced in 293T cells by co-transfection with the pSG3Δenv plasmid. The subtype B and C HIV-1 Reference Panels of Env Clones [49,50] were obtained through the NIH AIDS Research and Reference Reagent Program, Division of AIDS, NIAID, NIH, USA. The 96ZM651.02 gp160 clone was kindly provided by Dr D. Montefiori (Duke University Medical Center, Durham, NC) through the Comprehensive Antibody Vaccine Immune Monitoring Consortium (CA2 VIMC) as part of the Collaboration for AIDS Vaccine Discovery (CAVD). All additional pseudoviruses were produced at the VIMC laboratory.

#### 4.1.3. Cells

TZM-bl cells [49,51,52] were obtained through the NIH AIDS Research and Reference Reagent Program from J. C. Kappes, X. Wu, and Tranzyme, Inc., and cultured in Dulbecco’s modified Eagle medium (Invitrogen) containing 10% (*v*/*v*) fetal calf serum (FCS).

#### 4.1.4. VHH

A12, C8 and D7 were described by [15], and the 3D structure of D7 (highly homologous to A12) has been determined by Hinz et al. [53] A12 has been analyzed with EM tomography in complex with trimeric spikes [54]. 2E7, 1F10 and 1B5 are described in reference [17]. Selection of 1H9, 2B4F, 11F1F, 11F1B were not described earlier in detail (manuscript in preparation), J3 and 3E3 have previously been described [16,17,21]. VHH were purified as described previously [16,17].

### 4.2. Cross-Competition Assay

To be able to detect only one of the two VHH that are present during this assay, part of them had to be biotinylated. NHS-LC-LC-biotin (Thermo scientific, Cat. No: 21343), was incubated 10:1 with VHH, 1 h, RT. Unbound biotin was removed by dialysis. Biotinylated VHH (bVHH) were titrated.

MaxiSorp plates were coated with 100 ng/well gp140UG37 or gp140CN54 (for J3 250 ng gp140CN54) or 250 ng for gp120IIIB 250. After blocking, competing (non-biotinylated) VHH was added for 1 h. As control, binding of all competing VHH was detected separately. Subsequently 10 µL b-VHH was added to the competing VHH in concentration determined previously. B-VHH was detected by horseradish peroxidase (HRP) conjugated Streptavidin and visualized by o-Phenylenediamine, supplemented with 0.03% H_2_O_2_. Reaction was stopped using 1 M H_2_SO_4_ and signals measured at 490 nm. These values were then converted to percentages, in which competition with itself was regarded as maximal competition, i.e., 0%, and the competition against an irrelevant VHH as the unhindered binding, i.e., 100%.

### 4.3. Binding to Various Env Proteins

MaxiSorp plates were coated o/n with 100 ng gp140UG37, gp120IIIB or gp41. After blocking VHH (mono or bivalent) were added and detected by mouse anti Myc (9E10) and peroxidase conjugated donkey anti mouse (DAMPO). Visualization occurred as stated above.

### 4.4. Competition Assay with mAbs and sCD4

MaxiSorp plates were coated o/n with 100 ng gp140UG37 (for b12, 17b, sCD4 and partially 2F5 competition) or gp41 (4E10 and partially 2F5 competition). After blocking, VHH (mono or bivalent) were added and allowed to bind for 1 h, RT. 10 µL of competitor was added to the wells in a final concentration of 0.4 µg/mL b12, 1 µg/mL 17b, 0.07 µg/mL 4E10, 0.07 µg/mL 2F5 (gp140UG37) or 0.6 µg/mL 2F5 (gp41). Competitors mAbs b12, 17b, 2F5 and 4E10 were detected with peroxidase conjugated goat anti human (Jackson Immunoresearch), sCD4 by L120 and peroxidase conjugated donkey anti mouse. Visualization occurred as stated above.

For the titrated competition assays, MaxiSorp plates were coated overnight at 4 °C with 200 ng per well gp140UG37 (for 3E3, 1H9 and 1F10 competition) or gp41 (2E7 and 11F1F competition). Following blocking, the binding of HRP conjugated streptavidin to biotinylated VHH was detected using TMB-ELISA substrate (Pierce).

### 4.5. Construction of bi-Specific VHH

Essentially the procedure described by Hultberg 2011 has been followed, in short N- and C-terminal VHH fragments were amplified from their expression vectors, by PCR using DreamTaq green (Fermentas). Primers were used that annealed just outside of the VHH coding region and would add a linker and restriction sites to allow cloning of the N and C terminal VHH respectively. The resulting PCR products were purified using the NucleoSpin^®^ Extract II kit (Machery-Nagel, Düren, Germany), restricted with the appropriate enzymes and cloned into the VHH expression vector. Bacteria were transformed with the constructs by heatshock and subsequently clones were picked for sequence analysis. Bi-specific VHH were expressed in bacteria and purified by metal affinity chromatography in the same way as the monovalent VHH, which has been described before [16,17].

### 4.6. HIV Neutralization Assay

The HIV-1 neutralizing activities of the VHH were assessed in the TZM-bl cell based assay, as described previously [15,16,17]. No virus inactivation was observed with a negative control VHH. Cells were lysed with Bright-Glo luciferase reagent (Promega, Madison, WI, USA) and the luminescence measured. Fifty % inhibitory concentration (IC50) titers were calculated using the XLFit4 software (ID Business Solutions, Guildford, UK).

### 4.7. Epitope Mapping

The binding of VHH to arrays of peptides was assessed in a pepscan-based ELISA. Each well in the card contained covalently linked peptides that were incubated overnight at 4 °C with VHH, at a concentration of approximately 1 µg/mL, in PBS supplemented with 1% BSA and 0.1% Tween 80. After washing, the plates were incubated with a mouse anti-Histidine followed by HRP linked Rabbit anti-mouse (Southern Biotech, Birmingham, AL, USA) for 1 h at 25 °C. After further washing, peroxidase activity was assessed with an ABTS based substrate. The color development was quantified after 60 min using a charge-coupled device camera and an image-processing system.

### 4.8. Crystal Structure of VHH 2E7 in Complex with a gp41 Peptide

Purified VHH 2E7 was incubated with a 1.5 M excess of the gp41 peptide 582-AVERYLKDQQLLGIW-596 and crystallized at a concentration of 2.5 mg/mL. Crystals were obtained by the vapor diffusion method in sitting drops, with equal volumes of protein and reservoir solution (in 0.1M Hepes pH 7.0, 30% PEG 6000). The crystal was cryo-cooled at 100 K in absence of surrounding liquid [55]. A complete dataset was collected at the ESRF (Grenoble, France) beamline ID29. Data were processed and scaled with MOSFLM [56], and SCALA [57]. The crystals belong to space group P2_1_2_1_2_1_ with unit cell dimensions of a = 37.95 Å, b = 121.26 Å, c = 132.21 Å and three copies of the 2E7-gp41 peptide complex. The structure was solved by molecular replacement using PHASER [58] and the VHH structure from Protein Data Bank (PDB) ID 3EZJ as a search model. An initial model was built with ARP/wARP [59] and completed by several cycles of manual model building with Coot [60] and refinement with Refmac [61] using data to 2.9 Å resolution. The final model contains 2E7 residues 1–120 and gp41 residues 582 to 596. The R and R_free_ of the refined model are 19.5 and 24.6, respectively with 99% of the residues in the allowed regions of a Ramachandran plot. Molecular graphics figures were generated with PyMOL (W. Delano; http://www.pymol.org). Coordinates and structure factures have been deposited in the Protein Data Bank with accessions code 5HM1 (2E7).

### 4.9. SPR Analysis of 2E7 Binding to gp41_int_

The fusion intermediate conformation of gp41, gp41int was purified as described before [30]. Surface plasmon resonance (SPR) analysis was performed with a Biacore 3000 (GE Healthcare). As a flow buffer 10 mM HEPES, 150 mM NaCl, pH 7.4 with 0.005% Tween-20 was used. Gp41_int_ was immobilized to ~1200 response units using 50 µg/mL protein in flow buffer on an activated CM-5 sensor chip (GE Healthcare: BR-1000-50) according to the manufacturer’s instructions. Specific binding to the target protein was corrected for nonspecific binding to the deactivated control channel. The flow rate was 50 µL/min. Regeneration of the sensor chip was achieved with 10mM HCl followed by 4 M MgCl_2_ for 60 s at 60 µL/min. Data were analyzed with the BIA evaluation software version 4.1 and globally fit to t a 1:1 Langmuir model double referenced by subtraction of the blank surface and a blank injection.

## 5. Conclusions

Using various techniques we selected VHH against 4 different epitopes on HIV 1 gp140. This enabled us to rationally design bi-specific VHH. We show that increased potencies and extended breadths can be achieved by bi-specific VHH targeting two independent epitopes most likely on the same trimer, thereby producing important synergistic effects. Due to the lower risk of development of escape mutants and an improved efficacy, bi-specific anti-HIV-1 VHH may have great advantages over monovalent VHH in preventing HIV-1 transmission, either in a topical microbicide or when expressed from a gene therapy vector. Moreover, the lower production costs and higher stability of bi-specific VHH, make them superior to conventional antibodies. Because of its breadth and potency, VHH J3 was tested against SHIVs in a macaque challenge study quite successfully (manuscript in preparation).

At present the mono- and bispecific VHH are tested on their capability to recognize and destroy immune cells infected by HIV-1. Furthermore, an AAV based vector expressing VHH may provide prevention against infection and additional effects might be achieved by adding human Fc fragments to bi-specific VHH to suppress viremia, as was shown recently for the human antibody 3BNC117 [62].

## Figures and Tables

**Figure 1 antibodies-08-00038-f001:**
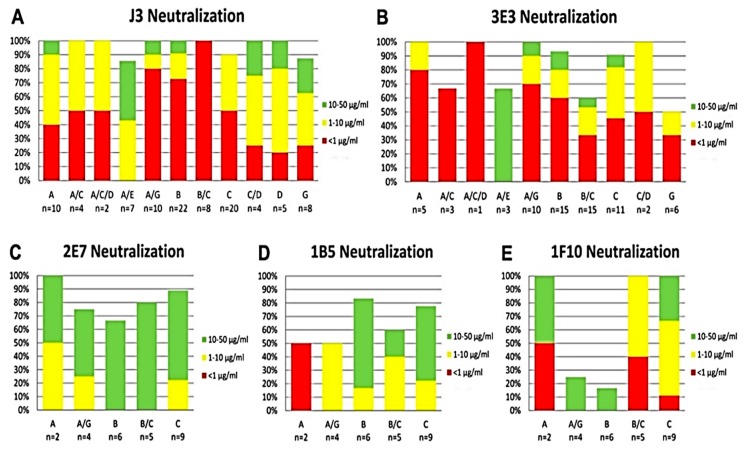
Clade Specific Neutralization of VHH. (**A**) J3, (**B**) 3E3, (**C**) 2E7, (**D**) 1B5 and (**E**) 1F10. The total neutralization per clade is shown by the height of the bar in the graph and the neutralization potency by the colors of the bar. Red indicates an IC50 < 1 µg/mL, yellow between 1 and 10 µg/mL and green between 10 and 50 µg/mL.

**Figure 2 antibodies-08-00038-f002:**
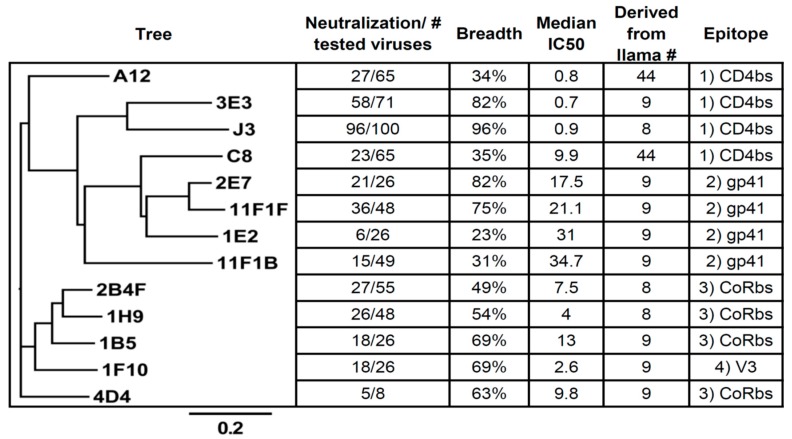
Phylogenetic Tree and Characteristics of the Selected VHH. VHH A12 (and D7) and C8 originate from an immunization with gp120 in llama 44, where the other VHH orginate from immunizations of llamas 8 and 9 with gp140 of UG037 and CN54. The neutralization (breadth), Medium IC50 values, epitopes (this study).

**Figure 3 antibodies-08-00038-f003:**
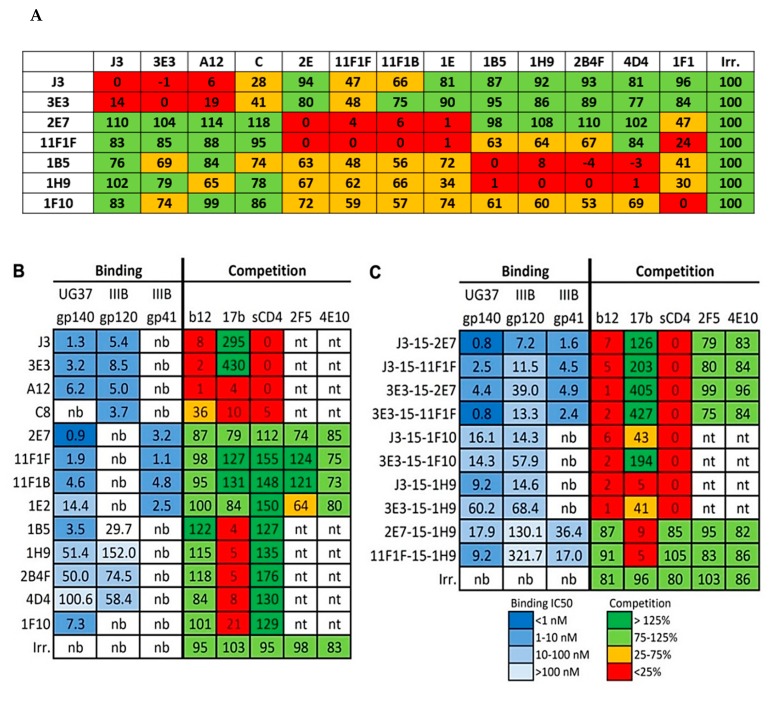
Interaction of the selected VHH with HIV-1 Env proteins. (**A**) Is showing the VHH competition among themselves for binding to gp140UG37, this is presented as a percentage. Gp140UG37 trimers used in this study are non-native trimers that contain a substantial amount of “open” Env structures. The signal observed for VHH competing with themselves was defined as 0%, signal during competition with an irrelevant VHH was defined as 100%. The VHH represented in the columns are the competing VHH, which were present in large excess. The VHH represented in the rows are the detected VHH. (**B**,**C**) are showing the binding of VHH (**B**) and bi-valent VHH (**C**) to gp140UG37, gp120IIIB and gp41, and their ability to compete with several mAbs and sCD4. Binding is expressed as the IC50 value in nM, competition as a percentage of the signal obtained where no competing VHH was present.

**Figure 4 antibodies-08-00038-f004:**
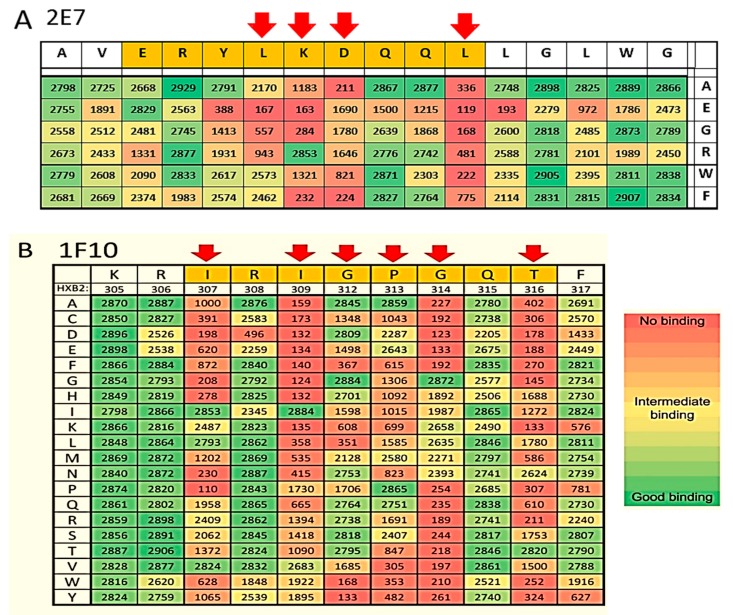
Epitopes Determination of the VHH 2E7, 11F1F and 1F10. After initial pepscan analyses, the amino acids in the gp140 envelop protein that interact with these VHH have been fine mapped by limited substitution of the amino acids of the epitopes of 2E7 (**A**) and full substitution for 1F10 (**B**). The key residues of the interaction are indicated by red arrows.

**Figure 5 antibodies-08-00038-f005:**
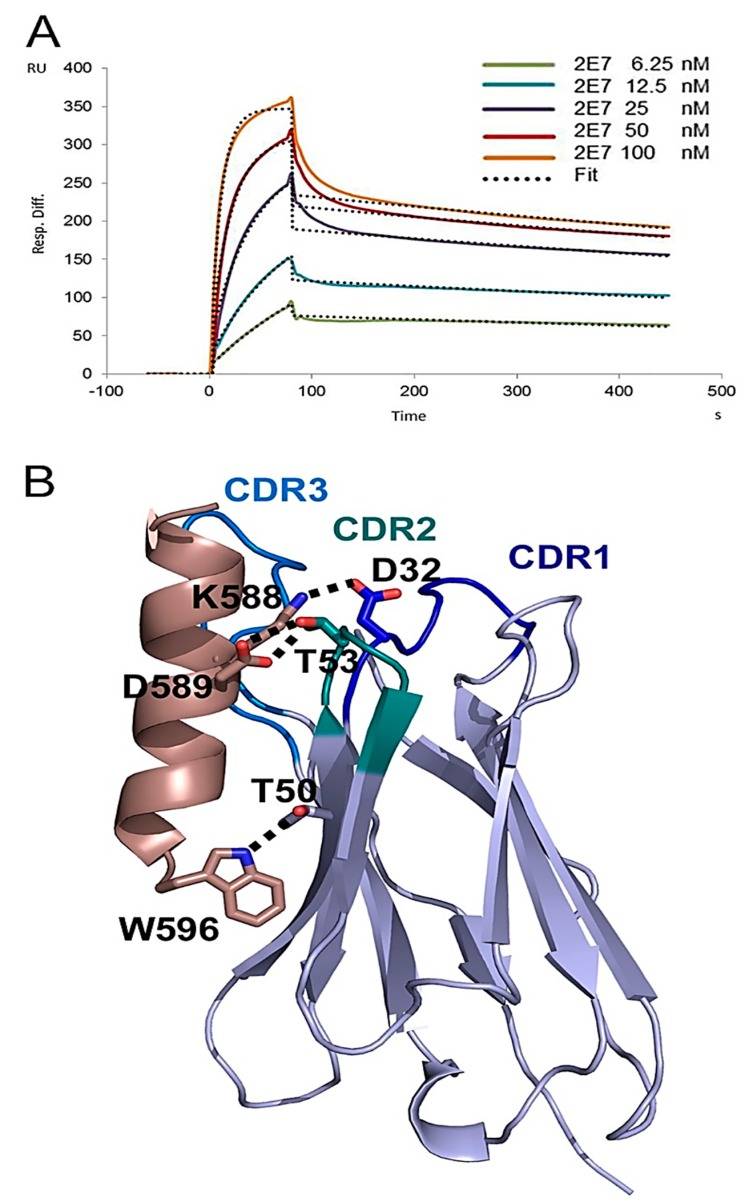
Structural analysis of 2E7 in complex with its gp41 HR1 epitope. (**A**) SPR analysis of VHH 2E7 with gp41int. 2E7 concentrations ranging from 6.25 to 100 nM were tested for binding to gp41int. Kinetic constants were derived by global fitting the data corresponding to the five indicated concentrations to a 1:1 Langmuir model (dotted lines) using local Rmax parameters. (**B**)Ribbon diagram of the crystal structure of the 2E7 VHH in complex with a gp41 HR1 peptide. The gp41 peptide is shown in wheat and the VHH CDR regions are indicated by different colors. Polar interactions are shown as dashed lines.

**Figure 6 antibodies-08-00038-f006:**
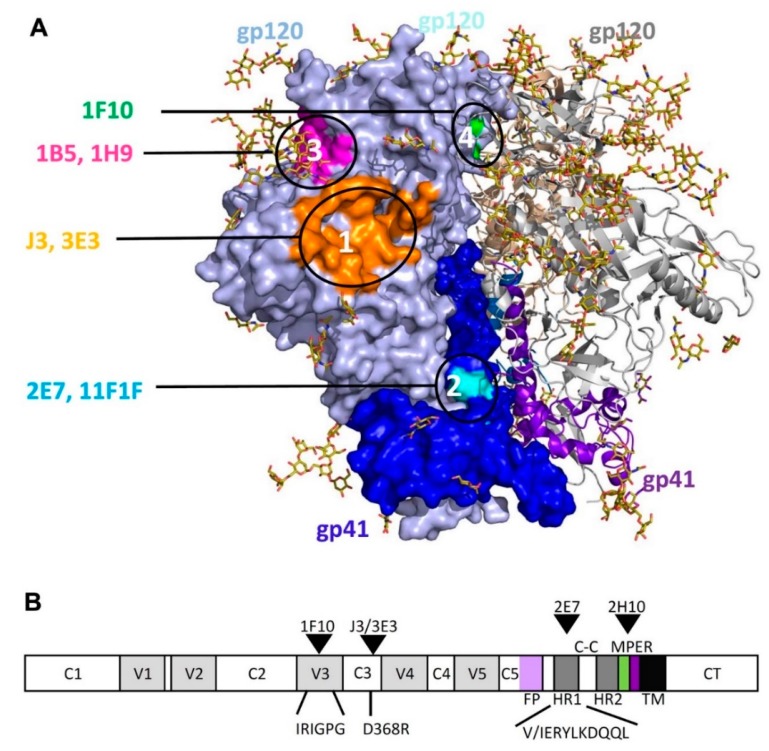
Mapping of the VHH epitopes onto the HIV-1 BG505 trimer (PDB ID 4TVP). (**A**) One monomer of the trimer is shown as a space filling model, carbohydrates are shown as sticks. The positions of the different VHH epitopes determined by crystallography and/or pepscan mapping are show in different colors. The distances between the epitopes were estimated and used to determine the linker length between VHH recognizing CD4bs (J3 and 3E3 and the epitopes recognized by the other VHH in the construction of a set of bi-specific VHH. The gp41 residues contacted by 2E7 are shown in Cyan, 1F10 in green. The 2E7 epitope located on gp41 HR1 is hidden in the trimer of the native Env gp140 conformation and not accessible for 2E7 binding. (**B**) Schematic of the domain organization of Env and location of the epitopes.

**Figure 7 antibodies-08-00038-f007:**
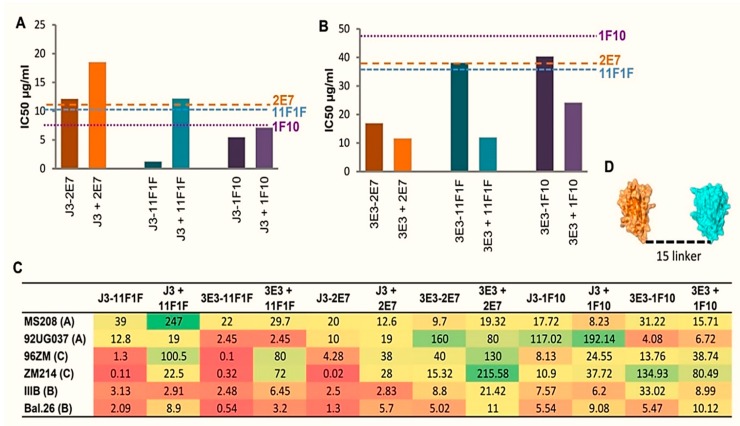
Broad and Potent HIV Neutralization by bi-specific VHH. (**A**,**B**) IC50 values in µg/ml for the indicated bi-specific VHH in comparison with the unlinked component VHH against viruses ((**A**) Du172.17 or (**B**) TV1.2) that were resistant to one of the VHH. Dotted lines represent the IC50 (μg/mL) for each component VHH and are color-coded in line with the legend. (**C**) IC50 nM for the indicated bi-specific VHH and the unlinked component VHH against the viruses indicated. IC50 values were generated from duplicate titrations of VHH onto TZM-bl cells as described in the materials and methods. Schematic representations of J3-2E7 bi-specific VHH (**D**).

**Table 1 antibodies-08-00038-t001:** Data collection and refinement statistics of 2E7GP41.

	2E7GP41	
Data Collection	Native	Anisotropic Scaling **
Wavelength (Å)	0.97239	
Space group	*P22_1_2_1_*	
Cell dimensions *a*, *b*, *c* (Å)	37.95, 121.26, 132.21	
Resolution (Å) *	44.68–2.95 (3.03–2.95)	44.68–2.95 (3.03–2.95)
Unique reflections *	13440 (993)	12538 (327)
R_merge_ (%)	8.4 (76.4)	7.9 (34.2)
*I/σI* *	13.8 (2.3)	14.6 (3.9)
Completeness (%) *	99.0 (100.00)	96.6 (34.5)
Redundancy *	4.7 (5.0)	4.3 (1.3)
Wilson *B* factor (Å^2^)	63.85	58.0
**Refinement**		
Resolution		44.68–2.96 (3.07–2.96)
*R_work_/R_free_* (%) *		19.42 (30.07)/24.91 (34.4)
**No. atoms**		
Protein		3087
Water		0
***B* factors (Å^2^)**		
Protein		55.4
Water		0
**r.m.s. deviations**		
Bond lengths (Å)		0.006
Bond angles (°)		1.26
**Ramachandran**		
Favored (%)		99.0
Outliers (%)		0.0
Clashscore *** Molprobity score ***		3.78 (100th percentile) 1.47 (100th percentile)

* Values in parentheses refer to highest resolution shell; ** The data were truncated according to the Diffraction Anisotropy Server of UCLA to 2.95 Å, 3.1 Å and 2.95 Å along *a*, *b*, *c*, respectively. M. Strong, M.R. Sawaya, S. Wang, M. Phillips, D. Cascio, D. Eisenberg, Proc. Natl. Acad. Sci. USA. 103, 8060-8-65, 2006. *** Percentiles are indicated for resolution range 2.962 Å ± 0.25 Å based on analysis with Molprobity.

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
