# Peer review of "Super Potent Bispecific Llama VHH Antibodies Neutralize HIV via a Combination of gp41 and gp120 Epitopes"

_2073-4468, 2019, doi:10.3390/antib8020038_

Round 1
Reviewer 1 Report
The manuscript by Strokappe et.al describes several new potent HIV neutralizing VHHs or Nanobodies. Moreover, some bi-specific constructs of these VHHs were shown to exhibit synergistic potency with neutralization capacities superior to simple mixture of the parent VHHs. Authors also solved two crystal structures of VHHs targeting two distinct epitopes. Structural data from these structures supports the epitope mapping by pepscan-based assay. Although, generally the data seems solid and interesting, the manuscript in its current state is very raw. Numerous typos, incomplete sentences, font differences, undeleted author notes make the text extremely hard to read. In addition, crystal structures are not sufficiently described and no data collection and refinement statistics are absent. These are essential for the evaluation of data quality.
Major comments:
1. Interaction of 1B5 VHH with gp120 is not described sufficiently. What was the resolution? Based on the available crystal structure does this VHH interact with the V3 loop? Are there any conformational changes in gp120 upon Nanobody binding? Data collection and refinement statistics have to be provided.
2. Fig 3 shows 2E7 binds unactivated pd140 - Please provide explanation, how is that possible considering that the crystal structure indicates 2E7 target epitope occluded in native ENvV conformation?
3. Fig 6 shows V3 loop is truncated – could the authors elaborate if the electron density for this loop was absent or the protein construct itself did not contain the loop?
4. Fig 7 It would be useful to mark the CD4 binding site
5. Please include these data described in line 300 as a lot of follow up descriptions are based on the data that is not shown.
6. Materials and methods section for the construction of bi-specific VHH should provide more information.
Minor comments:
1) Line 41 move ‘(VHH)’ after ‘antibodies’
2) Line 42 ‘as they can be produced relatively cheaply in microorganism like yeast ‘ please change to microorganisms like yeast or bacteria
3) Line 44 Font difference
4) Line 56 : 3E3 VHH was not introduced before.
5) Line 59-60 The sentence is grammatically incorrect.
6) Line 74 ‘potency, for various application’: Should be ‘applications’
7) Line 87 typo
8) Line 99 Please explain what is 17b
9) Line 114 Incomplete sentence
10) Line 118 Incomplete sentence
11) Line 130 ‘competition experiments of these VHH with sCD4 and b12 pointed out are really binding’ Word/words missing
12) Lines 529-533 – Authors forgot to delete their notes?? Font is different
Author Response
Reviewer 1:
The manuscript by Strokappe et.al describes several new potent HIV neutralizing VHHs or Nanobodies. Moreover, some bi-specific constructs of these VHHs were shown to exhibit synergistic potency with neutralization capacities superior to simple mixture of the parent VHHs. Authors also solved two crystal structures of VHHs targeting two distinct epitopes. Structural data from these structures supports the epitope mapping by pepscan-based assay. Although, generally the data seems solid and interesting, the manuscript in its current state is very raw. Numerous typos, incomplete sentences, font differences, undeleted author notes make the text extremely hard to read. In addition, crystal structures are not sufficiently described and no data collection and refinement statistics are absent. These are essential for the evaluation of data quality.
Major comments:
1. Interaction of 1B5 VHH with gp120 is not described sufficiently. What was the resolution? Based on the available crystal structure does this VHH interact with the V3 loop? Are there any conformational changes in gp120 upon Nanobody binding? Data collection and refinement statistics have to be provided.
We prefer not to show the data on gp120 binding to the VHH at this stage.
2. Fig 3 shows 2E7 binds unactivated pd140 - Please provide explanation, how is that possible considering that the crystal structure indicates 2E7 target epitope occluded in native ENvV conformation?
The UG37 trimers used in the binding experiments were non-native open trimers. Such trimers expose otherwise hidden gp41 epitopes. We have edited the text accordingly.
3. Fig 6 shows V3 loop is truncated – could the authors elaborate if the electron density for this loop was absent or the protein construct itself did not contain the loop?
The data has been taken out of the manuscript.
4. Fig 7 It would be useful to mark the CD4 binding site
In Figure 6A [present numbering] the CD4/J3 binding site is given. Soon the group op P. Kwong will publish details on the interaction based on co-crystallization. We have seen the data and it is clear that for over 95 % the interaction of J3 with CD4bs overlaps with that of CD4. Moreover it is also clear that J3 do no show interactions with amino acids outside the CD4bs of HIV-1
5. Please include these data described in line 300 as a lot of follow up descriptions are based on the data that is not shown.
Line 300 is referring to Figure 3A
6. Materials and methods section for the construction of bi-specific VHH should provide more information.
More detailed description added
Minor comments:
1) Line 41 move ‘(VHH)’ after ‘antibodies’ Done
2) Line 42 ‘as they can be produced relatively cheaply in microorganism like yeast ‘ please change to microorganisms like yeast or bacteria Done
3) Line 44 Font difference Done
4) Line 56 : 3E3 VHH was not introduced before. Done
5) Line 59-60 The sentence is grammatically incorrect. Done
6) Line 74 ‘potency, for various application’: Should be ‘applications’ Done
7) Line 87 typo corrected
8) Line 99 Please explain what is 17b Done
9) Line 114 Incomplete sentence Done
10) Line 118 Incomplete sentence Done
11) Line 130 ‘competition experiments of these VHH with sCD4 and b12 pointed out are really binding Line is completed
12) Lines 529-533 – Authors forgot to delete their notes?? Font is different Note has been deleted

Reviewer 2 Report
In this study, the authors employed a diverse set of approaches and characterized the epitope of 13 new llama single domain antibodies (VHH). These 13 VHH were further classified into four groups including CD4 binding site, gp41, co-receptor site
In this study, the authors employed a diverse set of approaches and characterized the epitope of 13 new llama single domain antibodies (VHH). These 13 VHH were further classified into four groups including CD4 binding site, gp41, co-receptor site, and V3 loop. Bi-specific VHH were constructed with J3 or 3E3 (CD4bs) in a combination with VHH from an independent epitope. An improved potencies and breadths were observed for certain bi-specific constructs when tested against selected HIV strains. This well-performed study offers novel insights into knowledge in the field.
While commendable, this study suffers from a few drawbacks. The competition studies and the neutralization characterization of bi-specific VHH are lacking in rigor. In addition, there are several statements made in the text that are either not correct, or not supported by the cited figures. Figure visualization and caption could be improved for better interpretability. Overall, this is a sound manuscript and I have several major and minor comments that should be addressed before considering acceptance.
Major comments:
(1) It is less clear to readers, why a different number of viral strains were tested against selected 13 VHH. Even for VHH that share the same epitope, the number of tested viruses varies greatly. In addition, for VHH 4D4, only 8 viruses were tested in neutralization assay, and neutralization breadth and median IC50 could be less meaningful in such case.
(2) A set of monoclonal antibodies (mAbs) were chosen for the competition study in Figure 3B. However, the competitive study is usually affected by the respective affinity. Otherwise one-way competition effects between mAbs and test VHH might lead to misinterpretation. Authors might consider providing additional data to exclude such a possibility.
(3) In Figure 8A&B, only linked J3-11F1F has demonstrated improved neutralization capacity when tested against Du172.17. However, no replicates and statistical analyses were included for these two figures. In addition, the rationality of the selection of an additional six viruses remains vague to readers. Are they resistant to the J3 or 3E3? What is the IC50 value for the component monomer? The author might want to test an expanded panel of the virus to justify “broad neutralization”
Minor comments:
(1) In the introduction, the authors stated that “none of the bi-specific antibodies comprising….showed improvement beyond the physical mixture of the component antibodies”. This is not true to the best of my knowledge. David Ho and colleagues (Huang et al., Cell 165), and Jeffrey Ravetch and colleagues (Bournazos et al., Cell 165) have engineered bi-specific antibodies/Fabs with improved activity.
(2) Authors might consider describing VHH 3E3 with a few sentences in the introduction.
(3) Authors stated that “an additional competition was performed with the gp41 binding mAbs 2F5 and 4E10……..do not compete with 2F5 and 4E10. Therefore we decided to determine their exact epitope on gp41”. This description is misleading and authors should consider mentioning that 2F5 and E410 are MPER binding antibodies
(4) For Figure 2, the meaning of Llama column is not clear to readers.
(5) Words are missing in the title of Figure 3.
(6) In Figure 4, the results of 11F1F (Figure 4B) is missing.
(7) In Figure 5, SPR measurements should be Figure 5A and crystal structure of the 2E7 VHH should be Figure 5B.
(8) Figure 6C does not have x-axis and y-axis labels. Moreover, are different concentrations of 1B5 tested in the SPR assay? If so, a color legend should be provided.
(9) In Materials and Methods 4.4 VHH, references were not cited in a consistent style.
(10) The manuscript could be improved with more details in “4.8 Construction of bi-specific VHH”. It is unclear to readers how bi-specific VHH were expressed and purified after constructs were cloned into expression vectors.
(11) The meaning of line 529 to line 533 is ambiguous to readers.
(12) In supplementary Figure 2C&D, the binding curves of control mAbs are missing.
Author Response
Reviewer 2 :
In this study, the authors employed a diverse set of approaches and characterized the epitope of 13 new llama single domain antibodies (VHH). These 13 VHH were further classified into four groups including CD4 binding site, gp41, co-receptor site
In this study, the authors employed a diverse set of approaches and characterized the epitope of 13 new llama single domain antibodies (VHH). These 13 VHH were further classified into four groups including CD4 binding site, gp41, co-receptor site, and V3 loop. Bi-specific VHH were constructed with J3 or 3E3 (CD4bs) in a combination with VHH from an independent epitope. An improved potencies and breadths were observed for certain bi-specific constructs when tested against selected HIV strains. This well-performed study offers novel insights into knowledge in the field.
While commendable, this study suffers from a few drawbacks. The competition studies and the neutralization characterization of bi-specific VHH are lacking in rigor. In addition, there are several statements made in the text that are either not correct, or not supported by the cited figures. Figure visualization and caption could be improved for better interpretability. Overall, this is a sound manuscript and I have several major and minor comments that should be addressed before considering acceptance.
Major comments:
(1) It is less clear to readers, why a different number of viral strains were tested against selected 13 VHH. Even for VHH that share the same epitope, the number of tested viruses varies greatly. In addition, for VHH 4D4, only 8 viruses were tested in neutralization assay, and neutralization breadth and median IC50 could be less meaningful in such case.
We agree that the VHH have been tested against different sets of viruses at different times. However, this does not affect the conclusions of the manuscript.
(2) A set of monoclonal antibodies (mAbs) were chosen for the competition study in Figure 3B. However, the competitive study is usually affected by the respective affinity. Otherwise one-way competition effects between mAbs and test VHH might lead to misinterpretation. Authors might consider providing additional data to exclude such a possibility.
We are aware of this, and in this particular assay there are even more variables to consider. Env is subjected to conformational change, so the affinity of a VHH or mAb can be affected by the binding of its competitor as well. In addition, competition does not even necessarily mean that they bind the same epitope, it may also show competition due to a conformational change. For the VHH we did test the competition both ways, for the mAbs we did not, as the detection method would be different, which could have also affect the interpretation.
However, our conclusions on epitope recognition are not solely based on the mAb competition. We use different techniques including pepscan analyses and crystallography to determine the epitopes. Although the interaction between 1B5 and pg120 is deleted from this manuscript, these data supports the competition studies. So in fact we have for all 4 groups of VHH the results of competition assays, for 3 out of 4 co-crystals studies supports our competition studies and 3 out of 4 pepscans does this as well.
(3) In Figure 8A&B, only linked J3-11F1F has demonstrated improved neutralization capacity when tested against Du172.17. However, no replicates and statistical analyses were included for these two figures. In addition, the rationality of the selection of an additional six viruses remains vague to readers. Are they resistant to the J3 or 3E3? What is the IC50 value for the component monomer? The author might want to test an expanded panel of the virus to justify “broad neutralization”
J3 does not neutralize Du172.17 (and 3E3 not TV1.2) so it is expected to see no improvement. However, we show here that the linkage of J3 to the other VHH does not destroy their capacity to neutralize Du172.17. The other 6 strains are selected to have 2 more strains from clade A, B and C and was based on the rational to include more clade A, B and C viruses. The IC50 values of most monomer VHH were determined previously
Minor comments:
(1) In the introduction, the authors stated that “none of the bi-specific antibodies comprising….showed improvement beyond the physical mixture of the component antibodies”. This is not true to the best of my knowledge. David Ho and colleagues (Huang et al., Cell 165), and Jeffrey Ravetch and colleagues (Bournazos et al., Cell 165) have engineered bi-specific antibodies/Fabs with improved activity.
We appreciate the comment and have changed the text and references accordingly.
(2) Authors might consider describing VHH 3E3 with a few sentences in the introduction.
Done
(3) Authors stated that “an additional competition was performed with the gp41 binding mAbs 2F5 and 4E10……..do not compete with 2F5 and 4E10. Therefore we decided to determine their exact epitope on gp41”. This description is misleading and authors should consider mentioning that 2F5 and E410 are MPER binding antibodies.
Manuscript has been adjusted
(4) For Figure 2, the meaning of Llama column is not clear to readers.
Figure has been adjusted to make this more clear
(5) Words are missing in the title of Figure 3.
Title of figure 3 has been amended.
(6) In Figure 4, the results of 11F1F (Figure 4B) is missing.
Figure 4 has been updated.
(7) In Figure 5, SPR measurements should be Figure 5A and crystal structure of the 2E7 VHH should be Figure 5B.
Figure 5 has been updated.
(8) Figure 6C does not have x-axis and y-axis labels. Moreover, are different concentrations of 1B5 tested in the SPR assay? If so, a color legend should be provided.
The data presented in the previous figure 6 has been removed.
(9) In Materials and Methods 4.4 VHH, references were not cited in a consistent style.
This inconsistency has been adjusted.
(10) The manuscript could be improved with more details in “4.8 Construction of bi-specific VHH”. It is unclear to readers how bi-specific VHH were expressed and purified after constructs were cloned into expression vectors.
More details have been added to this section.
(11) The meaning of line 529 to line 533 is ambiguous to readers.
(12) In supplementary Figure 2C&D, the binding curves of control mAbs are missing.
Reviewer 3 Report
The authors present a interesting paper to develop a bsVHH that has increased potency for HIV. Overall the results support the conclusions drawn, however, there are deficiencies in presentation of the manuscript provided as is.
Major Comments
Structures:
Please supply the PDB deposition validation reports for both structures to be published.
Update PDB deposition code to be a complete code for 2E7, supply the deposition code for 1B5.
Table of statistics for refinement etc is required in either the MS or the SI to allow assessment of the model validation - There is only a reference to table 00201 on line 188.
Section 2.2, 2.3, 2.4, 2.5 should be considered as subsections of section 2.1 (epitope mapping).
In particular
Section 2.2 contains reference to no results within this MS (either in the body or SI) rather refers to an additional MS in prep and to data that is not presented and is more of a discussion point.
Figure 4. title is not informative 'determination of the VHH2E7, 11F1F and 1F10.
Figure 5: A and B of figures and legend do not match
The data in figure 5 appears to be used for modelling of kinetics, no fits are supplied or indication of the goodness of fit. The data has large disturbances after inj stop. it is not clear if the the data is double reference subtracted. The methodology described in 4.12 (line 518-522) is insufficient particularly compared to the detailed explanation provided of the experiment on the T200 (lines 523-528)
Figure 6:
A. V3 loop (truncated) - is this loop truely truncated or just not modelled into the density?
C. SPR KD: 0.64 +/- 0.02 nM affinity determined using a 400, 200, 100, 50nM concentration series. The derived affinity is well below the range tested. Ideally concentration rage for affinity determination would be 10X below and 10X above KD for affinity (what is Rmax for this data). In the corresponding methods there is no information on the goodness of fits, nor n=x times data carried out
Data presented in figures is not linked to the text/results including:
No reference to Figure 1 in the text
No reference to figure 2 in the text
No reference to SI-fig2 in the text
Line 499-512: it is unclear what complex was solved HXBc2:1B5 or 93TH057:1B5
Methods:
4.8 - is poorly written, brief and does not describe VHH production.
4.11 - is not sufficient. The information provided for 2E7 is absent for 1B5 (space group, dimensions etc). This data should be provided as a table of statistics. No refinement data for 1B5 complex.
There is no corresponding description of how the IC50 assay was carried out.
Typographical
Line 227: change Kd to KD and nm to nM
Figure 6 change Kd to KD
Line 430: change streptavidine to streptavidin
Line 529, 530, 531, 532, 533 - correct or add the required information.
Author Response
Reviewer 3:
The authors present a interesting paper to develop a bsVHH that has increased potency for HIV. Overall the results support the conclusions drawn, however, there are deficiencies in presentation of the manuscript provided as is.
Major Comments
Structures:
Please supply the PDB deposition validation reports for both structures to be published.
Update PDB deposition code to be a complete code for 2E7, supply the deposition code for 1B5.
Answer: The 2E7 code has been amended and the validation report is being provided.
As indicated above the data referring to the structure of 1B5 in complex with gp120 has been taken out of the manuscript.
Table of statistics for refinement etc is required in either the MS or the SI to allow assessment of the model validation - There is only a reference to table 00201 on line 188.
Table 1, describing the crystallographic statistics of 2E7 in complex with the gp41 peptide has been added to the manuscript.
Section 2.2, 2.3, 2.4, 2.5 should be considered as subsections of section 2.1 (epitope mapping).
In particular
Section 2.2 contains reference to no results within this MS (either in the body or SI) rather refers to an additional MS in prep and to data that is not presented and is more of a discussion point.
Remarks to this remarks on statistics. We realize that the provided information does not contain statistics. However that was not the purpose of this study. For a long time it was thought that in non primate animals no broadly neutralizing antibodies against HIV could be raised. We proofed that this was wrong. Moreover we proofed that with appropriate selection and competition methods out of the hundreds of VHH, VHH can be found that recognize clearly different epitopes on gp140 and that one of the selected VHH, J3 gives excellent protection in a macaque challenge trial [will be published]
Figure 4. title is not informative 'determination of the VHH2E7, 11F1F and 1F10.
Title has been adjusted
Figure 5: A and B of figures and legend do not match
This has been corrected.
The data in figure 5 appears to be used for modelling of kinetics, no fits are supplied or indication of the goodness of fit. The data has large disturbances after inj stop. it is not clear if the the data is double reference subtracted. The methodology described in 4.12 (line 518-522) is insufficient particularly compared to the detailed explanation provided of the experiment on the T200 (lines 523-528)
We now show the fitting of the 2E7 SPR data together with the experimental data in figure 5A. The data is double reference subtracted as indicated in the methods section.
Figure 6:
A. V3 loop (truncated) - is this loop truely truncated or just not modelled into the density?
C. SPR KD: 0.64 +/- 0.02 nM affinity determined using a 400, 200, 100, 50nM concentration series. The derived affinity is well below the range tested. Ideally concentration rage for affinity determination would be 10X below and 10X above KD for affinity (what is Rmax for this data). In the corresponding methods there is no information on the goodness of fits, nor n=x times data carried out
Figure 6 has been removed from the manuscript.
Data presented in figures is not linked to the text/results including:
No reference to Figure 1 in the text
No reference to figure 2 in the text
No reference to SI-fig2 in the text
Line 499-512: it is unclear what complex was solved HXBc2:1B5 or 93TH057:1B5
1B5 structure has been removed from the manuscript.
In the revised text we have addressed these comments
Methods:
4.8 - is poorly written, brief and does not describe VHH production.
More details have been given on the construction and production of the bi-specific VHH.
4.11 - is not sufficient. The information provided for 2E7 is absent for 1B5 (space group, dimensions etc). This data should be provided as a table of statistics. No refinement data for 1B5 complex.
1B5 structure has been removed from the manuscript.
There is no corresponding description of how the IC50 assay was carried out.
Typographical
Line 227: change Kd to KD and nm to nM Adjusted
Figure 6 change Kd to KD Figure taken out
Line 430: change streptavidine to streptavidin Adjusted
Line 529, 530, 531, 532, 533 - correct or add the required information. Done
Round 2
Reviewer 1 Report
Most comments were addressed by taking the structural data out of the manuscript. However, since there are still biophysical data supporting the conclusions of the manuscript it doesn't not change the overall merit.